# RNA Interference Analysis of the Functions of Cyclin B in Male Reproductive Development of the Oriental River Prawn (*Macrobrachium nipponense*)

**DOI:** 10.3390/genes13112079

**Published:** 2022-11-09

**Authors:** Wenyi Zhang, Pengchao Wang, Yiwei Xiong, Tianyong Chen, Sufei Jiang, Hui Qiao, Yongsheng Gong, Yan Wu, Shubo Jin, Hongtuo Fu

**Affiliations:** 1Key Laboratory of Freshwater Fisheries and Germplasm Resources Utilization, Ministry of Agriculture and Rural Affairs, Freshwater Fisheries Research Center, Chinese Academy of Fishery Sciences, Wuxi 214081, China; 2Wuxi Fisheries College, Nanjing Agricultural University, Wuxi 214081, China; 3National Demonstration Center for Experimental Fisheries Science Education, Shanghai Ocean University, Shanghai 201306, China

**Keywords:** *Macrobrachium nipponense*, cyclin B, RNAi analysis, insulin-like androgenic gland hormone, spermatogenesis

## Abstract

*Cyclin B* (*CycB*) plays essential roles in cell proliferation and promotes gonad development in many crustaceans. The goal of this study was to investigate the regulatory roles of this gene in the reproductive development of male oriental river prawns (*Macrobrachium nipponense*). A phylo-genetic tree analysis revealed that the protein sequence of *Mn*-*CycB* was most closely related to those of freshwater prawns, whereas the evolutionary distance from crabs was much longer. A quantitative PCR analysis showed that the expression of *Mn*-*CycB* was highest in the gonad of both male and female prawns compared to that in other tissues (*p* < 0.05), indicating that this gene may play essential roles in the regulation of both testis and ovary development in *M. nipponense*. In males, *Mn*-*CycB* expression in the testis and androgenic gland was higher during the reproductive season than during the non-reproductive season (*p* < 0.05), implying that *CycB* plays essential roles in the reproductive development of male *M. nipponense*. An RNA interference analysis revealed that the *Mn-insulin-like androgenic gland hormone* expression decreased as the *Mn-CycB* expression decreased, and that few sperm were detected 14 days after the *dsCycB* treatment, indicating that *CycB* positively affects testis development in *M. nipponense*. The results of this study highlight the functions of *CycB* in *M*. *nipponense*, and they can be applied to studies of male reproductive development in other crustacean species.

## 1. Introduction

The oriental river prawn (*Macrobrachium nipponense*) (Crustacea; Decapoda; Palaemonidae) is an economically important freshwater aquaculture species in China [1], with an annual production that reached 225,321 tons in 2019 [2]. In this species, gonad maturity occurs approximately 40 days after hatching in both male and female prawns [3]. This rapid gonad development leads to inbreeding between young prawns, resulting in mating and the propagation of multiple generations in the same ponds, which, in turn, negatively affects the market size of prawns [4,5]. Therefore, an artificial technique to extend the period of testis development in *M. nipponense* is urgently needed.

The X-organ–sinus gland complex, which is located in the eyestalk of crustaceans, is a principal neuroendocrine complex that stores and releases many neurosecretory hormones [6]. These hormones play vital roles in regulating many biological processes in crustacean species [7,8,9,10,11,12,13,14,15,16,17], and some of them have negative regulatory effects on the development of both the testis and ovary in *M. nipponense*. For example, the RNA interference (RNAi) of the *gonad-inhibiting hormone* expression promoted ovarian development [18], and the eyestalk ablation of male prawns stimulated the expression of the *Mn-insulin-like androgenic gland hormone* (*Mn-IAG*) and promoted testis development [4,5]. A similar phenomenon was reported for whiteleg shrimp (*Litopenaeus vannamei*) [17] and Chinese mitten crabs (*Eriocheir sinensis*) [19]. Thus, the eyestalk–androgenic gland–testis endocrine axis in males has important regulatory effects on male sexual differentiation and reproduction in crustaceans [20,21]. A transcriptome profiling analysis of the androgenic gland after the eyestalk ablation of male *M. nipponense* revealed that the cell cycle was the main metabolic pathway enriched in differentially expressed genes [4]. This suggests that the cell cycle is involved in the regulation of reproductive development in male *M. nipponense* [22,23]. *Cyclin B* (*CycB*) was enriched in the cell cycle metabolic pathway, suggesting that this gene may promote the reproductive development of male *M. nipponense* [4].

Gametogenesis is an important part of gonad development in multi-cellular organisms. Several cell cycle regulators play essential roles in gametogenesis including cyclins, cyclin-dependent kinase (CDK), and the cyclin-dependent kinase inhibitor [24]. The maturation promotion factor (MPF), a key regulator of cell proliferation in eukaryotic cells, can stimulate both mitotic and meiotic cell cycles [25].The MPF is a heterodimer composed of CycB protein and CDK1 [26,27,28]. CycB protein abundance fluctuates periodically during cell proliferation, which is required to activate or inhibit MPF activity [29]. This process is strongly related to the cell cycle and to the activation of CDK1. *CycB* was reported to be involved in oogenesis in crustacean species, such as *Eriocheir sinensis* [30], *Penaeus monodon* [31,32], *Marsupenaeus japonicus* [33], *Scylla paramamosain* [34,35], and *Metapenaeus affinis* [36], and in spermatogenesis in *S. paramamosain* [34].

The goal of the present study was to analyze the potential functions of *CycB* in the male reproductive development of *M*. *nipponense*. The results of this study highlight the regulatory functions of *CycB* in male *M*. *nipponense* and can be used to develop an artificial technique to regulate the process of testis development in this species.

## 2. Methods and Materials

### 2.1. Ethics Statement

We obtained permission from the Institutional Animal Care and Use Ethics Committee of the Freshwater Fisheries Research Center, Chinese Academy of Fishery Sciences (Wuxi, China) to conduct all experiments involving *M. nipponense*. All *M*. *nipponense* used in the present study were collected from the Dapu *M. nipponense* Breeding Base in Wuxi, China (120°13′44″ E, 31°28′22″ N). All prawns were maintained in aerated freshwater for 3 days before the tissue collection. The dissolved oxygen content in the water was maintained at ≥6 mg/L. Prawns were anesthetized using an ice bath prior to sampling.

### 2.2. Rapid Amplification of cDNA Ends (RACE)

RNA from the testis of *M*. *nipponense* was extracted to synthesize the template for 3′cDNA and 5′cDNA cloning. Well-described procedures for RACE cloning and the identification of sequence characteristics were followed [37,38]. Table 1 lists the specific primers used for the *Mn-CycB* cloning, and Table 2 lists the accession numbers of the protein sequences of *Cdc2*s from different species. These protein sequences were used to measure the evolutionary distance of *Cdc2* between different species via the construction of a phylo-genetic tree using MEGA X. The phylo-genetic tree was constructed using the maximum likelihood method and bootstrap method with 1000 replications.

### 2.3. Quantitative PCR (qPCR) Analysis

qPCR was used to measure the relative mRNA expression of *Mn-CycB* in tissue samples (eyestalk, brain, heart, hepatopancreas, muscle, gonad, and gill) collected as shown in Figure 1. Each tissue was collected from five different prawns and pooled to form a biological replicate; three biological replicates were analyzed for each tissue. All collected tissues were immediately preserved in liquid nitrogen until they were used for the qPCR analysis. Previously reported methods for RNA isolation and cDNA synthesis were followed [37,38]. Table 1 lists the primers used for the qPCR analysis. The eukaryotic translation initiation factor 5A is a stable reference gene for the PCR analysis in *M. nipponense*, and it was used to normalize the *Mn-CycB* expressions in the present study [39]. The relative mRNA expressions of *Mn-CycB* were calculated using the 2^−∆∆CT^ comparative CT method [40]. Data are shown as the mean ± standard deviation (SD) of tissues from three replicates.

### 2.4. RNAi Analysis

The potential function of *CycB* in the reproductive development of male *M. nipponense* was investigated with RNAi. Snap Dragon (http://www.flyrnai.org/cgibin/RNAifind_primers.pl, accessed on 12 March 2021) was used to design the specific RNAi primer with a T7 promoter site based on the open reading frame of *Mn-CycB* (Table 1). *Mn-CycB* dsRNA (*dsCycB*) and green fluorescent protein dsRNA (*dsGFP*) were synthesized using the Transcript Aid™ T7 High Yield Transcription Kit (Fermentas, Waltham, MA, USA) following the manufacturer’s protocol. *dsGFP* was used as the negative control [41].

Six hundred male *M*. *nipponense* (3.14−4.32 g) were collected approximately 5 months after hatching and randomly divided into the *dsCycB*-treated group (RNAi) and *dsGFP*-treated group (control). The injected dose of *dsCycB* or *dsGFP* was 4 μg/g [42,43]. Seven days after the first injection, prawns were injected with 4 μg/g of *dsCycB* or *dsGFP*. Androgenic gland samples were collected from both groups on days 1, 7, and 14 after the first injection of *dsGFP* or *dsCycB*. The tissue collection and qPCR analysis procedures were the same as those described in Section 2.3. The *Mn-CycB* and *Mn-IAG* mRNA expressions were measured with qPCR using the same cDNA templates to assess the regulatory relationship between *CycB* and *IAG* in *M. nipponense*.

### 2.5. Hematoxylin and Eosin (HE) Staining

The morphological differences in the testis between *dsGFP*-treated and *dsCycB*-treated prawns were evaluated using the histological observations of tissue sections stained with HE following well-described protocols [44,45]. Briefly, the tissues were embedded in paraffin and sliced into 5 µm thick sections using a microtome (Leica, Wetzlar, Germany). The sectioned tissues were placed on a slide and stained with HE for 3–8 min. The slides were observed and photos were taken under an Olympus SZX16 microscope (Tokyo, Japan).

### 2.6. Statistical Analysis

All statistical analyses were conducted using SPSS Statistics 23.0 (IBM, Armonk, NY, USA). The Shapiro–Wilk test and Bartlett test were used to measure the normality and homogeneity of variances, respectively. Independent sample *t*-tests were used to assess the statistical differences in *Mn-CycB* and *Mn-IAG* expression between *dsCycB*-treated and *dsGFP*-treated prawns on the same day, *Mn-CycB* expressions in the same tissues between male and female prawns, and *Mn-CycB* expressions in the testis and androgenic gland between the reproductive and non-reproductive seasons. Statistical differences in different mature tissues and different developmental stages were identified with the analysis of variance (ANOVA), followed by Duncan’s multiple range test. Quantitative data were expressed as mean ± SD. A *p*-value < 0.05 was considered to be statistically significant.

## 3. Results

### 3.1. Mn-CycB cDNA Sequence Analysis

The full cDNA sequence of *Mn-CycB* was 1862 base pairs (bp) long with an open reading frame of 1197 bp encoding 398 amino acids. The 5′ and 3′ un-translated regions were 177 bp and 488 bp, respectively (Figure 2). The cDNA sequence of *Mn-CycB* was submitted to the NCBI with the accession number OP379746. The theoretical isoelectric point and the molecular weight of the *Mn-CycB* protein were 9.17 and 45.172 kDa, respectively.

A BLASTX analysis in the NCBI revealed that the similarities between the nucleotide sequence of *Mn-CycB* and the *CycB*s in other freshwater prawns were >80%, including *Macrobrachium rosenbergii* (95.99%), *Palaemon modestus* (85.71%), and *Palaemon carinicauda* (84.46%) (Figure 3). The deduced protein sequence of *Mn-CycB* contained some highly conserved sites or domains, including the cyclin destruction box (RxALGxIxN), which is highly conserved among the known B-type cyclins. It is located at amino acids (aa) 35–43 and regulates cyclin destruction via the ubiquitin proteasome pathway in the N-terminal region. A conserved pkA site (RRxSK) located at 266–270 aa was also found, which is characteristic of B-type cyclins (Figure 3).

### 3.2. Phylo-Genetic Tree Analysis

Well-defined *CycB* sequences were identified using the BLASTX analysis in the NCBI, and then a condensed phylo-genetic tree based on the amino acid sequence of *Mn-CycB* and these other *CycB* sequences was generated using MEGA X. The amino acid sequence of *Mn-CycB* first clustered with the amino acid sequences of freshwater prawns as a group, and then it clustered with marine shrimp as a big group. The evolutionary distance from crabs was much longer. *Mn-CycB* was most closely related to the *CycB* of *M. rosenbergii* (Figure 4).

### 3.3. Mn-CycB Expression Analysis

The physiological function of *Mn-CycB* in *M. nipponense* was preliminarily assessed by measuring its expression in various tissues using qPCR. *Mn-CycB* expression was highest in the ovary and testis of female and male *M. nipponense*, respectively. Its expression was higher in the eyestalk, muscle, gonad, and gill of female prawns compared to male prawns (*p* < 0.05), whereas the opposite expression pattern was found in the brain, heart, and hepatopancreas (*p* < 0.05) (Figure 5A).

*Mn-CycB* mRNA was widely expressed during the different developmental stages of *M. nipponense*. Its expression was higher during the embryonic developmental stages than during the larval and post-larval developmental stages. The highest expression of *Mn-CycB* mRNA occurred at the cleavage stage, when it was significantly higher than that of the other tested stages (*p* < 0.01) (Figure 5B). Additionally, *Mn-CycB* mRNA expression levels in the testis and androgenic gland were 2.34-fold and 1.97-fold higher, respectively, during the reproductive season than during the non-reproductive season (*p* < 0.01) (Figure 6).

### 3.4. RNAi Analysis

An RNAi analysis was employed to reveal the regulatory functions of *CycB* in the reproductive development of male *M. nipponense*. The expression of *Mn-CycB* on day 1 after the *dsGFP* injection was lower than the values on days 7 and 14 (*p* < 0.05), and the expression did not differ significantly between days 7 and 14 (*p* > 0.05). However, the injection with *dsCycB* significantly decreased *Mn-CycB* expression by approximately 55% on day 1 after treatment (*p* < 0.05), compared with the control group on the same day. The expression decreased by over 85% and 90% on days 7 and 14 after the *dsCycB* injection (*p* < 0.01), respectively (Figure 7A). The qPCR analysis also revealed that the *Mn-IAG* expression decreased by approximately 66%, 77%, and 80% on days 1, 7, and 14 after treatment with *dsCycB* (*p* < 0.01), respectively, compared with the control group on the same day (Figure 7B).

### 3.5. Histological Observations

Histological observations revealed that sperm cells were the main cell type (>50%) in the testis of *M. nipponense* on different days after the *dsGFP* injection, and they were much more abundant than spermatogonia and spermatocytes. The percentage of sperm did not differ significantly on different days in the control group. However, treatment with *dsCycB* significantly decreased the number of sperm cells over time, while the percentages of spermatogonia and spermatocytes increased. Sperm were rarely observed on day 14 after the *dsCycB* treatment, and the majority of cells were spermatogonia (Figure 8).

## 4. Discussion

*CycB* plays essential roles in the process of cell proliferation through the activation of CDK1 [29] and in the regulation of testis development in silk moths (*Bombyx mori*) [46] and *S. paramamosain* [34]. A previous transcriptome profiling analysis revealed that *Mn-CycB* expression was significantly up-regulated in the androgenic gland after the ablation of eyestalks from male *M. nipponense*. This finding and the negative regulatory relationship between the hormones secreted by eyestalks and male reproductive development in *M. nipponense* indicated that *CycB* may be involved in male development [4,5]. The goal of the present study was to investigate the potential functions of *CycB* in the regulation of reproductive development in male *M. nipponense*.

The BLASTX analysis showed that the similarity between *Mn-CycB* and the *CycB* of *M. rosenbergii* reached 95.99%, and *Mn-CycB* shared >80% identity with the *CycB*s of other freshwater prawns. The *Mn-CycB* amino acid sequence contained some highly conserved sites or domains of B-type cyclins, including the cyclin destruction box (RxALGxIxN) at 35–43 aa and the pkA site (RRxSK) at 266–270 aa, which was consistent with the results of previous studies [34,35]. The phylo-genetic tree analysis revealed that *Mn-CycB* had the closest evolutionary relationship with *CycB*s of freshwater prawns, and then with marine prawns, whereas the evolutionary relationship with crabs was much more distant. More protein sequences of *CycB*s from freshwater prawns need to be identified for the better evolutionary analysis of *Mn-CycB*.

Previous studies reported that the dominant expression site of *CycB* was the gonad in many aquatic animals. In female *S. paramamosain*, *CycB* expression was highest in the ovary [34,35]. In female *M. affinis, CycB* expression was highest in the ovary, followed by the muscle, thoracic ganglion, and heart [36]. *CycB* was predominantly expressed in the ovary and testis of *E. sinensis* [30], and in female *P. monodon,* expression was highest in the ovary [31]. *CycB* mRNA was also expressed in the testis, ovary, gill, mantle, muscle, and eggs of zebra mussels (*Dreissena polymorpha*) [47]. Similarly, the highest expression of *Mn-CycB* occurred in the testis of males and the ovary of females in the present study. In addition, the expression of *Mn-CycB* in the testis and androgenic gland was higher during the reproductive season than during the non-reproductive season. In *S. paramamosain*, *CycB* expression in the testis developmental stage two was significantly higher than in stages one and three, suggesting that *CycB* may play essential roles in the spermatogenesis of this species [34]. Histological observations revealed that testis and androgenic gland development during the reproductive season was more vigorous than that during the non-reproductive season in *M. nipponense* [48,49]. Furthermore, *Mn-CycB* mRNA was widely expressed during the different developmental stages of *M. nipponense*, but expression during the embryonic developmental stages, especially at the cleavage stage, was higher than the expression during larval and post-larval development. Overall, these results indicated that *CycB* plays essential roles in the process of the embryogenesis of *M. nipponense*.

RNAi has been used to investigate the regulatory functions of *CycB* in many aquatic species. For example, the knockdown of *CycB* expression in sea urchins delayed oocyte meiosis re-initiation [50]. Many RNAi studies have focused on the functions of *CycB* in ovarian development, but few have reported on the regulatory roles of *CycB* in male reproductive development. In the present study, the injection of *dsCycB* resulted in a significant decrease in both *Mn-CycB* and *Mn-IAG* expression in male *M. nipponense*, indicating that *CycB* has positive regulatory effects on the expression of *Mn*-*IAG*. The androgenic gland is a special organ in male crustaceans. The hormones secreted by the androgenic gland promote male sexual differentiation and reproduction, especially the development of testis and male sexual characteristics [51,52,53]. Male *M. rosenbergii* undergo sex reversal to become female when the androgenic gland is ablated [51,54,55]. *IAG* is the main gene expressed in the androgenic gland and it was reported to positively regulate male differentiation and reproductive development. This was illustrated through the knockdown of *IAG* using RNAi, which had a significant inhibitory effect on spermatogenesis in *M. rosenbergii* [56]. Similar functions of *IAG* have been reported in many other crustacean species [57,58,59,60,61]. In the current study, the relationship between the expression of *CycB* and *IAG* indicated that *CycB* was involved in the regulation of male reproductive development in *M. nipponense*. In the histological analysis, no significant differences in the testis were observed on different days after prawns were injected with *dsGFP*, and sperm were the dominant cell type (>50% of cells). However, the number of sperm cells in the testis decreased with time after the *dsCycB* treatment, and sperm cells were scarce on day 14 in the RNAi group. These results showed that *CycB* regulated testis development in *M. nipponense* by inhibiting the expression of *Mn-IAG*.

In conclusion, the data presented herein indicate that *CycB* positively regulates testis development and spermatogenesis by affecting the expression of *IAG* in *M. nipponense*. The results of this study highlight the functions of *CycB* in *M*. *nipponense*, and they can be applied to studies of male reproductive development in other crustacean species.

## Figures and Tables

**Figure 1 genes-13-02079-f001:**
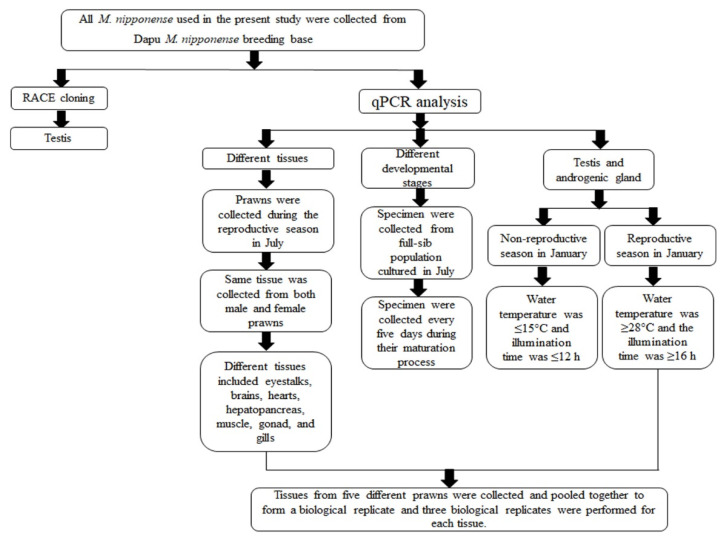
Flowchart showing the sample collection procedure.

**Figure 2 genes-13-02079-f002:**
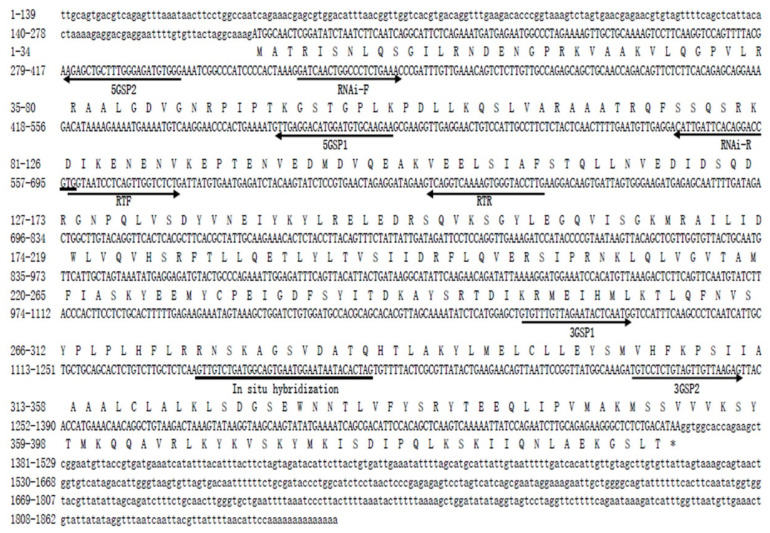
Nucleotide and deduced amino acid sequence of *Mn-CycB*. The nucleotide sequence is displayed in the 5′–3′ direction and is numbered on the left. The deduced amino acid sequence is shown as a single capital letter amino acid code. The 3′ and 5′ untranslated regions are shown with lowercase letters. Codons are numbered on the left, the methionine (ATG) initiation codon is shown, and an asterisk denotes the termination codon (TGA). Arrows and line indicated the direction of location of each primer.

**Figure 3 genes-13-02079-f003:**
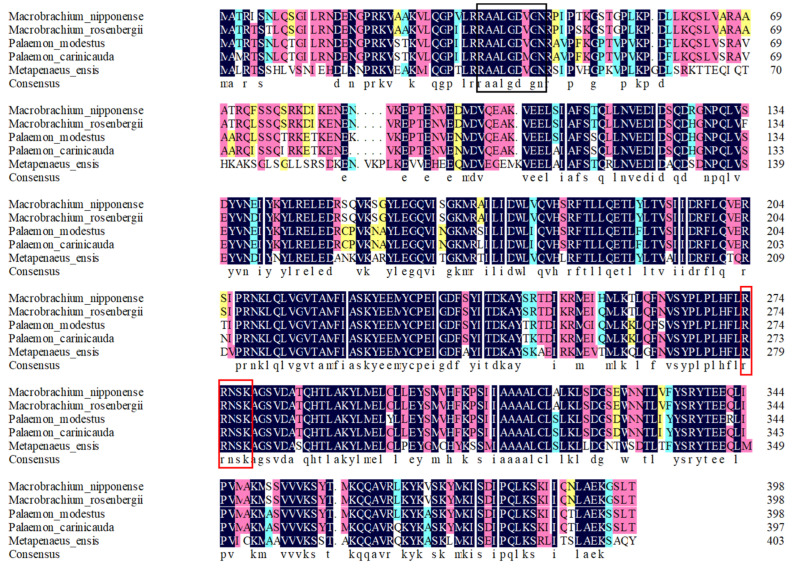
Similarities in amino acid sequences of *CycB* between different species. The black box indicates the cyclin destruction box and the red box indicates the conserved pkA site.

**Figure 4 genes-13-02079-f004:**
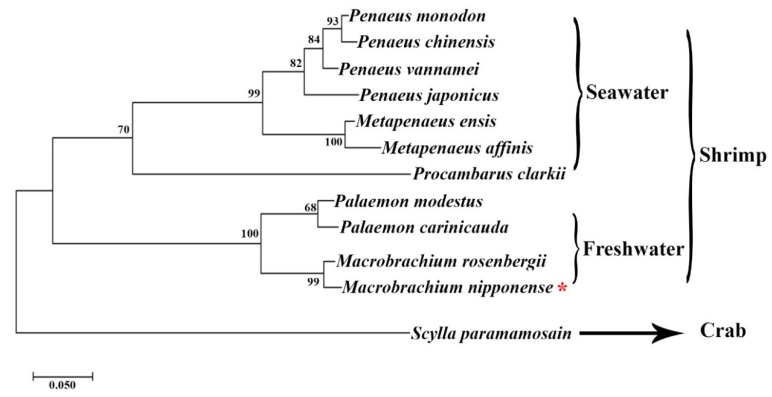
Phylo-genetic tree of *CycB*s from different organisms. Species names are listed on the right side of the tree. The red asterisk indicates *M. nipponense*.

**Figure 5 genes-13-02079-f005:**
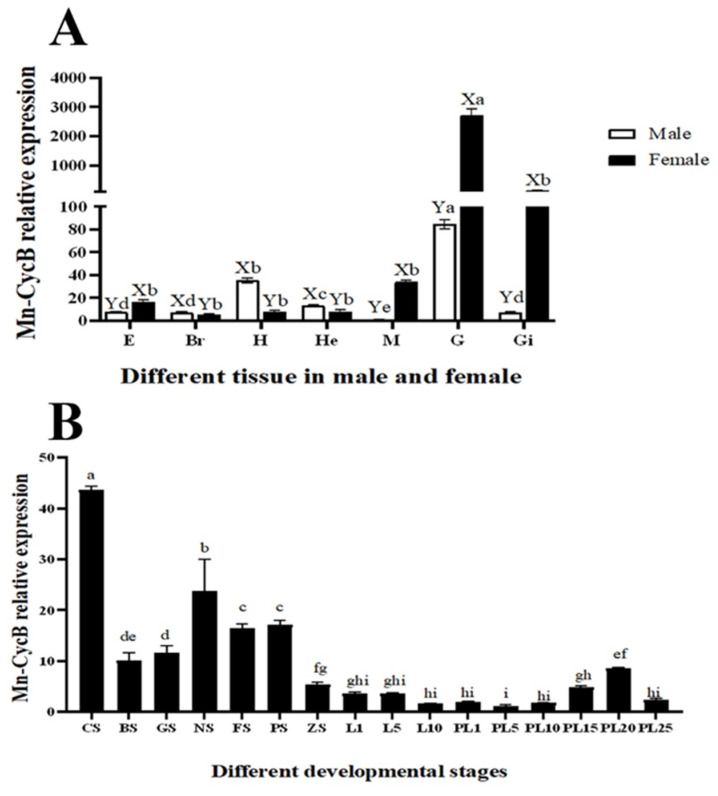
*Mn-CycB* expression in different mature tissues and developmental stages as determined with qPCR. Lowercase letters on the bars indicate expression differences between different tissue samples (analysis of variance (AVONA)). Capital letters on the bars indicate expression differences in the same tissue between male and female prawns (independent sample *t*-tests). (**A**) *Mn-CycB* expression in different mature tissues. (**B**) *Mn-CycB* expression in different developmental stages. E: eyestalk; Br: brain; H: heart; He: hepatopancreas; M: muscle; G: gonad; Gi: gill; CS: cleavage stage; BS: blastula stage, GS: gastrula stage; NS: nauplius stage; FS: posterior nauplius stage; PS: protozoa stage; ZS: zoea stage; L: larval developmental stage; PL: post-larval developmental stage.

**Figure 6 genes-13-02079-f006:**
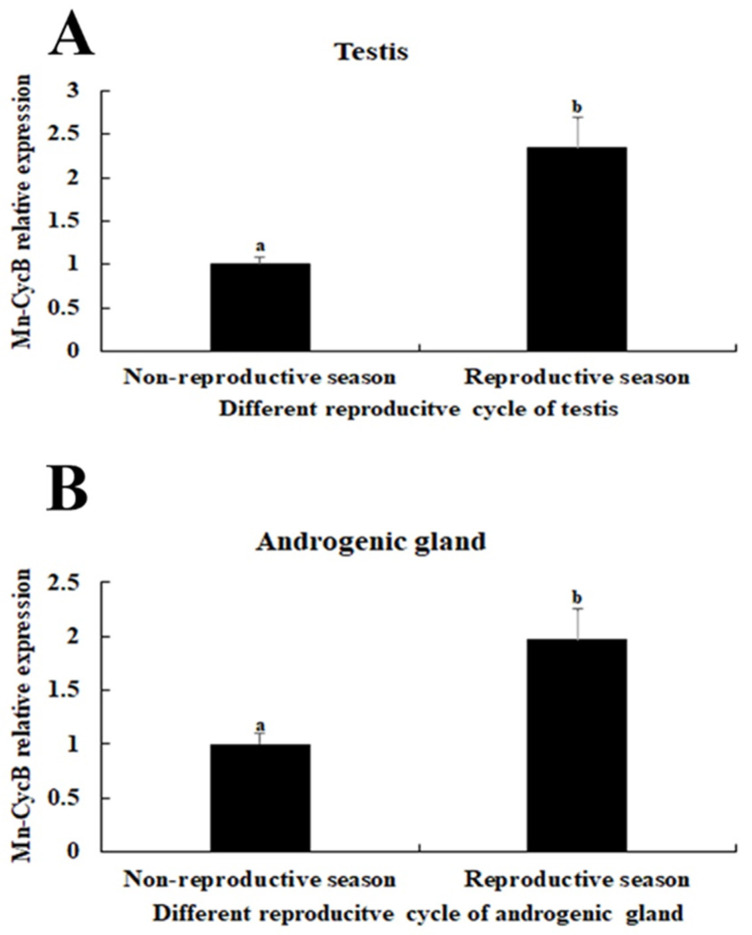
*Mn-CycB* expression in the testis and androgenic gland of prawns collected during the reproductive and non-reproductive seasons. Lowercase letters (a, b) on the bars indicate expression differences between different tissue samples (independent sample *t*-tests). (**A**) *Mn-CycB* expression in the testis. (**B**) *Mn-CycB* expression in the androgenic gland.

**Figure 7 genes-13-02079-f007:**
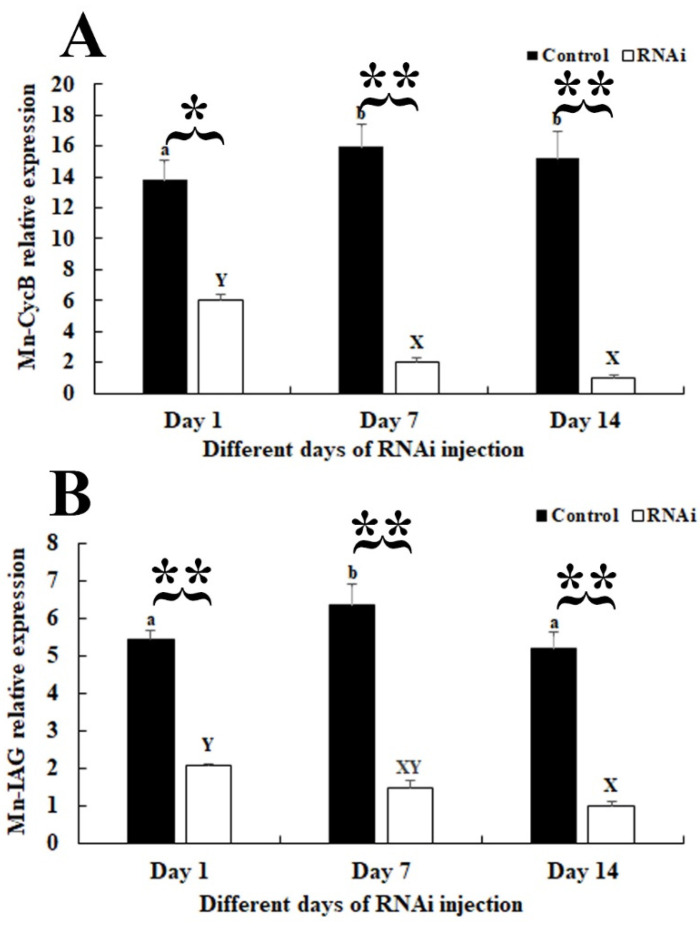
Measurement of *Mn-CycB* and *Mn-IAG* expression in the androgenic gland on different days after *dsCycB* and *dsGFP* injections. Lowercase letters (a, b) on the bars indicate expression differences between different days after *dsGFP* injection, and capital letters (X, Y) on the bars indicate expression differences between different days after *dsCycB* injection (analysis of variance (AVONA)). * *p* < 0.05 and ** *p* < 0.01 indicate significant expression differences between the *dsGFP*- and *dsCycB*-treated groups on the same day (independent sample *t*-tests). (**A**) *Mn-CycB* expression on different days after *dsGFP* and *dsCycB* injections. (**B**) Measurement of *Mn-IAG* expression on different days after *dsGFP* and *dsCycB* injections.

**Figure 8 genes-13-02079-f008:**
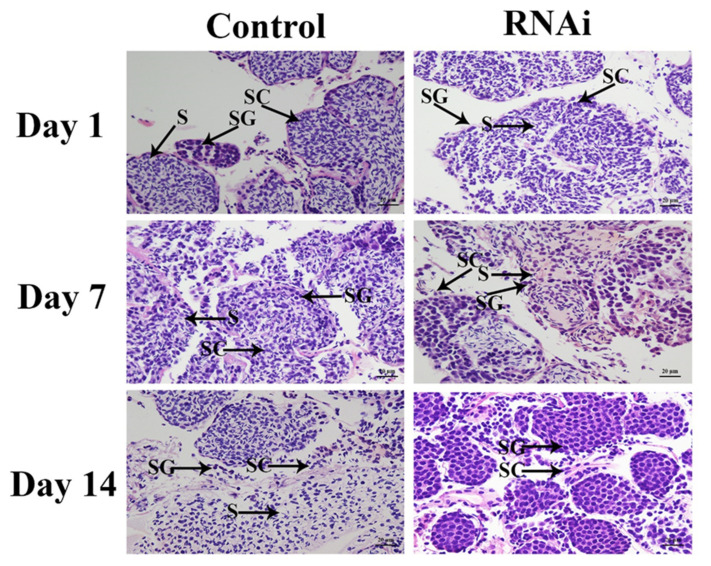
Histological comparison of testis tissue between *dsGFP*- and *dsCycB*-treated prawns. SG: spermatogonia; SC: spermatocyte; S: sperm. Scale bar = 20 μm.

**Table 1 genes-13-02079-t001:** Universal and specific primers used in this study.

Primer Name	Nucleotide Sequence (5′→3′)
CycB-3GSP1	GTGTTTGTTAGAATACTCAATG
CycB-3GSP2	TGTCCTCTGTAGTTGTTAAGAG
CycB-5GSP1	TTCTTGCACATCCATGTCCTCAA
CycB-5GSP2	CCACATCTCCCAAAGCAGCTCT
3′RACE OUT	TACCGTCGTTCCACTAGTGATTT
3′RACE IN	CGCGGATCCTCCACTAGTGATTTCACTATAGG
5′RACE OUT	CATGGCTACATGCTGACAGCCTA
5′RACE IN	CGCGGATCCACAGCCTACTGATGATCAGTCGATG
CycB-RTF	TGGTAATCCTCAGTTGGTCTCTG
CycB-RTR	CAAGGTACCCACTTTTGACCTGA
IAG-RTF	CGCCTCCGTCTGCCTGAGATAC
IAG-RTR	CCTCCTCCTCCACCTTCAATGC
EIF-F	CATGGATGTACCTGTGGTGAAAC
EIF-R	CTGTCAGCAGAAGGTCCTCATTA
CycB RNAi-F	TAATACGACTCACTATAGGGGATCAACTGGCCCTCTGAAA
CycB RNAi-R	TAATACGACTCACTATAGGGCACGGTCCTGTGAATCAATG

**Table 2 genes-13-02079-t002:** Protein sequences of *Cdc2* from different species used in this study.

Species	Accession Number
*Macrobrachium nipponense*	ADB44902.1
*Macrobrachium rosenbergii*	ADP95148.1
*Palaemon modestus*	QDE09442.1
*Palaemon carinicauda*	AKA66439.1
*Metapenaeus ensis*	ADI86225.1
*Metapenaeus affinis*	ADI86226.1
*Penaeus monodon*	ACH72072.1
*Penaeus japonicus*	AAV37462.1
*Procambarus clarkii*	ALD48736.1
*Penaeus vannamei*	XP_027209834.1
*Penaeus chinensis*	XP_047471868.1
*Scylla paramamosain*	ACN54752.1

## Data Availability

The data generated and analyzed during this study are included in this article.

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
