# Peer review of "RNA Interference Analysis of the Functions of Cyclin B in Male Reproductive Development of the Oriental River Prawn (Macrobrachium nipponense)"

_genes, 2022, doi:10.3390/genes13112079_

Round 1

Reviewer 1 Report

Genes

Identification of potential functions of Cyclin B in male sexual development of the oriental river prawn, Macrobrachium nipponense by RNAi analysis

General: It is a good article with important findings regarding the Cyclin B. The authors might want to change the title and instead of sexual development insert reproduction. So, the title might be "Identification of potential functions of Cyclin B in male reproduction of the oriental river prawn, Macrobrachium nipponense by RNAi analysis" since the study is dealing with adult males and not involved in any developmental stages. The weakness of the article is the use of adult animals, so it is not teaching us anything regarding the IAG-switch and sex differentiation.

The article is suggesting a major involvement of Cyclin B in the regulation of IAG.  However, IAG and the IAG-sexual switch are not mentioned in the introduction (e.g. Frontiers in Endocrinology 11:651 2020).  The Eyestalk - AG - testis endocrine axis is deeply described in other Macrobrachium species in the literature. The literature here is very shallow, limited to M. niponense.

The phenotype is not described correctly. The results (histology) are suggesting effects on sperm formation from the spermatozoa stage and not on proliferation and nothing on sexual development. This part should be completely rewritten describing the results more accurately.

Minor:

Line 1: Title: should be Identification, the I is missing

Line 20: " the evolutionary relationship with the crab was dramatically long" do the authors mean that the evolutionary distance is long?

Line 33: key words: instead of " Testis development" should be spermatogenesis since the article does not deal with development. Also might consider adding AG and IAG? "Histological observations" is not a good key word.

Introduction

line 50: “The X-organ–SG complex (XO–SG)” SG should be sinus gland in full when first mentioned.

Line 51: “which was considered as a principal neuroendocrine gland to store and release many neurosecretory hormones” The XO-SG is a complex, please replace ‘gland’ with complex.

Methods and materials:

Line 92: What about anesthetic measures during surgical interventions?

Line 96: RNA from Testis was extracted.

Line 114: Table 2 only the species. which gene in the title of the Table?

line 130:” Three biological replicates were performed for each tissue.” for qPCR, it is more reliable to perform the analysis with more replicates (at least 6).

line 160-161: “Androgenic gland samples were collected from both the control group and the RNAi group on days 1, 7, and 14, after dsGFP and dsCycB injection.” When? after the last injection? or the first injection? Please clarify.

Statistical Analysis- 

lines 179-180: “The paired t-test was used to measure the statistical difference between the control group and the RNAi group on the same day.” The treatment and control samples are from different animals and cannot be considered a pair. Therefore, paired t-test is not the appropriate statistical test.

lines 180-181: “Statistical differences were identified by ANOVA analysis of variance followed by least significant difference and Duncan’s multiple range tests.” According to the one-way ANOVA assumptions, did the author test the normality and homogeneity of variances before conducting the ANOVA test?

Results

Line 209 "cyclin destruction box" appears twice.

218: also here "evolutionary relationship" should be changed to "evolutionary distance".

Line 225 as well as the Figure 4 the title of 3.3 should be Mn-CycB expression and on the Y axis of Figure 4 it should be "Mn-CycB relative expression". When the calculation is based on 2−ΔΔCT as described in the methods part (line 144), the Y axis represents the relative transcript levels and not the expression levels, please clarify and correct.

In figure 4A the results with the gills of females is puzzling, suggesting again that 3 biological replicates are not sufficient.

Figure 4 caption (lines 241-250) – please add the relevant statistical tests for each part.

Figure 5: again, three biological replicates are too low. The Y axis is not clear. What kind of relativity? Is it %?

Figure 6: it is not clear in which tissue was this done. This should be specified in the caption and in the text. If it is the AG tissue why didn’t the authors include this tissue also in Figure 4A??

lines 266-268: “qPCR analysis revealed that the Mn-CycB expression generally remained at a stable level on different days after the treatment of dsGFP, although the expression of Mn-CycB at day 1 was slightly lower than those of day 7 and day 14 (< 0.05).” To which of the differences is the p value mentioned referring to? if it is referring to the difference between day 1 to 7 and 14, than it should be mentioned as statistically significant rather than “slightly”. In general it could be either significant or not.

Figure 6 – please add the relevant statistical tests for each part.

Figure 6 caption line 283: “Lowercases indicated expression difference between different days after dsGFP and dsCycB injection.” did the statistical analysis include all samples of both dsGFP and dsCycB injected groups together? or two separate analyses? if separately, why did the author use the same letters? please clarify and mark in different letter.

Discussion:

lines 332-334: “In the present study, the highest expressions of Mn-CycB were observed in the testis and ovary of male and female M. nipponense, respectively, which were consistent with the previous studies.” This statement does not represent the exact results, as shown in figure 4A. the transcript levels of female gill are higher than the testis. 

Lines 347-350: here the authors are presenting a misleading conclusion.  No functional genomics experiment was done in the period of P5-PL20. So they only could suggest that this should be tested. Or the authors could perform functional genomics at this stage.

Line 360: here again the literature is not touching the most closest Macrobrachium species AG and IAG literature and the IAG-switch.

 The final sentence:

In conclusion, the above evidence in the present study indicated that CycB positively  regulated the testis development  spermiogenesis through affecting the expression of IAG in M. nipponense. This study dramatically provides valuable evidences on the calls for further studies of CycB in  other crustacean species, as well as promoting the male sexual development.

Author Response

Reviewer 1

General: It is a good article with important findings regarding the Cyclin B. The authors might want to change the title and instead of sexual development insert reproduction. So, the title might be "Identification of potential functions of Cyclin B in male reproduction of the oriental river prawn, Macrobrachium nipponense by RNAi analysis" since the study is dealing with adult males and not involved in any developmental stages. The weakness of the article is the use of adult animals, so it is not teaching us anything regarding the IAG-switch and sex differentiation.

Reply: We agreed with your comment. According to your comment, we have revised the title as: RNAi analysis identified the Cyclin B functions in male reproduction of Macrobrachium nipponense.

As you mentioned that the weakness of the article is the use of adult animals, so it is not teaching us anything regarding the IAG-switch and sex differentiation. The prawn’s sizes of M. nipponense during the sex-sensitive period are only approximate 0.1 g. They are too small to perform the RNAi analysis during the sex-sensitive period of M. nipponense. Thus, in the present study, we only aimed to identify the regulatory roles of Cyclin B in male reproduction of M. nipponense. The functions of Cyclin B in the IAG-switch and sex differentiation will be performed by Cas9 when we break through this technique in M. nipponense.

The article is suggesting a major involvement of Cyclin B in the regulation of IAG.  However, IAG and the IAG-sexual switch are not mentioned in the introduction (e.g. Frontiers in Endocrinology 11:651 2020).  The Eyestalk - AG - testis endocrine axis is deeply described in other Macrobrachium species in the literature. The literature here is very shallow, limited to M. niponense.

Reply: Thank you for your suggestion. We agreed with your comment. Some additional information has been provided in the manuscript at line 63-71. We hope it is much better now.

The phenotype is not described correctly. The results (histology) are suggesting effects on sperm formation from the spermatozoa stage and not on proliferation and nothing on sexual development. This part should be completely rewritten describing the results more accurately.

 Replay: we agreed with your comment. The manuscript was mainly focused on the functions of the CycB in male reproduction of M. nipponense. We have revised it through the manuscript.

Minor:

Line 1: Title: should be Identification, the I is missing

Reply: Thank you for your suggestion. The correction has been done. Reviewer 2 suggested that the title is too long. Thus, revised the title as “RNAi analysis identified the Cyclin B functions in male reproduction of Macrobrachium nipponense” in the revised manuscript.

Line 20: " the evolutionary relationship with the crab was dramatically long" do the authors mean that the evolutionary distance is long?

Reply: Thank you for your suggestion. Yes, it is the evolutionary distance, we have revised it through the manuscript.

Line 33: key words: instead of " Testis development" should be spermatogenesis since the article does not deal with development. Also might consider adding AG and IAG? "Histological observations" is not a good key word.

Reply: Thank you for your suggestion. The key words were revised according to the comment.

Introduction

line 50: “The X-organ–SG complex (XO–SG)” SG should be sinus gland in full when first mentioned.

Reply: Thank you for reminding. The full name of “X-organ–SG complex” has been provided as “X-organ–sinus gland complex” at line 58 in the manuscript.

Line 51: “which was considered as a principal neuroendocrine gland to store and release many neurosecretory hormones” The XO-SG is a complex, please replace ‘gland’ with complex.

Reply: Thank you for your reminding. “Complex” has been used in the manuscript at line 59.

Methods and materials:

Line 92: What about anesthetic measures during surgical interventions?

Reply: Thank you for your suggestion. The anesthetic measures has been provided as “The sampling was performed under anesthetization with an ice bath” at line 103-104.

Line 96: RNA from Testis was extracted.

Reply: Thank you for your reminding. We revised this sentence as “RNA from testis was extracted to synthesize the template for 3′cDNA and 5′cDNA cloning” in the manuscript at line 106-107.

Line 114: Table 2 only the species. which gene in the title of the Table?

Reply: Thank you for your reminding. We have revised the title of Table as “Protein sequences of Cdc2 from different species used in this study”.

line 130:” Three biological replicates were performed for each tissue.” for qPCR, it is more reliable to perform the analysis with more replicates (at least 6).

Reply: Thank you for your suggestion. In the experiment, each biological replicate was randomly collected from five individuals, and three biological replicates included 15 prawns.  In addition, the expressions in each tissue are quite stable, and the standard deviation between these three biological replicates are reasonable. Thus, we believe the qPCR data in the present study are reliable. We agree with your comment that six biological replicates are more reliable for qPCR analysis. We will perform six biological replicates for qPCR analysis in our further experiment.

line 160-161: “Androgenic gland samples were collected from both the control group and the RNAi group on days 1, 7, and 14, after dsGFP and dsCycB injection.” When? after the last injection? or the first injection? Please clarify.

Reply: Thank you for your reminding. Actually, we began to collect the sample after the injection for the first time. Thus, the sample for day 1 and day 7 were collected after the injection for the first time, and then we injected the dsCdc2 and dsGFP for the second time in the left prawns, and the samples for day 14 were collected after the second injection. We clarify it as “The samples began to be collected after the injection for the first time” at line 160-161 in the manuscript.

Statistical Analysis- 

lines 179-180: “The paired t-test was used to measure the statistical difference between the control group and the RNAi group on the same day.” The treatment and control samples are from different animals and cannot be considered a pair. Therefore, paired t-test is not the appropriate statistical test.

Reply: Thank you for your reminding. Here, we used the independent samples T-test to compare the statistical difference between the control group and the RNAi group on the same day. We have revised it in Statistical Analysis at line 179-183.

lines 180-181: “Statistical differences were identified by ANOVA analysis of variance followed by least significant difference and Duncan’s multiple range tests.” According to the one-way ANOVA assumptions, did the author test the normality and homogeneity of variances before conducting the ANOVA test?

Reply:  Thank you for your comment. The Shapiro–Wilk test and Bartlett test were respectively used to measure the normality and homogeneity of variances. We also provided it in the manuscript at line 178-179.

Results

Line 209 "cyclin destruction box" appears twice.

Reply: Thank you for your reminding. One “cyclin destruction box” was deleted from the figure legend of figure 2.

218: also here "evolutionary relationship" should be changed to "evolutionary distance".

Reply: Thank you for your suggestion. We have revised the "evolutionary relationship" as "evolutionary distance" through the manuscript.

Line 225 as well as the Figure 4 the title of 3.3 should be Mn-CycB expression and on the Y axis of Figure 4 it should be "Mn-CycB relative expression". When the calculation is based on 2−ΔΔCT as described in the methods part (line 144), the Y axis represents the relative transcript levels and not the expression levels, please clarify and correct.

Reply: Thank you for your suggestion. We agreed with your suggestion. The explanation of Y axis in Figure 4, Figure 5 and Figure 6 has been revised according to your comment.

In figure 4A the results with the gills of females is puzzling, suggesting again that 3 biological replicates are not sufficient.

Reply: Thank you for your suggestion. In the experiment, each biological replicate was randomly collected from five individuals, and three biological replicates included 15 prawns.  In addition, the expressions in each tissue are quite stable, and the standard deviation between these three biological replicates are reasonable. Thus, we believe the qPCR data in the present study are reliable. We speculate that the expression of Mn-CycB has sex difference, and we will evaluate the phenomenon in future study.

Figure 4 caption (lines 241-250) – please add the relevant statistical tests for each part.

Reply: Thank you for your suggestion. The relevant statistical tests for each part have been added in the Figure 5-7 captions.

Figure 5: again, three biological replicates are too low. The Y axis is not clear. What kind of relativity? Is it %?

Reply: Thank you for your reminding. The Y axis is "Mn-CycB relative expression". We have revised it in the Figure 5. The standard deviation between these three biological replicates is reasonable. Thus, we believe the qPCR data in the present study is reliable.

 Figure 6: it is not clear in which tissue was this done. This should be specified in the caption and in the text. If it is the AG tissue why didn’t the authors include this tissue also in Figure 4A??

Reply: Yes, androgenic gland (AG) is collected to perform the qPCR analysis after the injection of dsCdc2 and dsGFP, as it mentioned in the Materials and methods section of line 169. We have clarified it in the figure legend of “Figure 6”. For figure 4-A, the same tissues were collected from both male and female prawns. However, AG did not exist in female prawns of M. nipponense. Thus, AG did not include in figure 4-A. The current time is non-reproductive season for M. nipponense. Thus, the data for qPCR analysis in AG cannot be supplied now. We will determine the Mn-CycB expression in VG in our further studies.

lines 266-268: “qPCR analysis revealed that the Mn-CycB expression generally remained at a stable level on different days after the treatment of dsGFP, although the expression of Mn-CycB at day 1 was slightly lower than those of day 7 and day 14 (< 0.05).” To which of the differences is the p value mentioned referring to? if it is referring to the difference between day 1 to 7 and 14, than it should be mentioned as statistically significant rather than “slightly”. In general it could be either significant or not.

Reply: Thank you for your reminding. We agreed with your comment. We have revised this part, according to the statistical analysis. The revision can be seen as “qPCR analysis revealed that the expression of Mn-CycB at day 1 after dsGFP injection was lower than those of day 7 and day 14 (p < 0.05), while the Mn-CycB expressions between day 7 and day 14 showed no significant difference (p > 0.05)” at line 234-237.

Figure 6 – please add the relevant statistical tests for each part.

Reply: The relevant statistical tests for each part have been added.

Figure 6 caption line 283: “Lowercases indicated expression difference between different days after dsGFP and dsCycB injection.” did the statistical analysis include all samples of both dsGFP and dsCycB injected groups together? or two separate analyses? if separately, why did the author use the same letters? please clarify and mark in different letter.

Reply: Thank you for your reminding. The statistical analysis was performed separately between the two groups. According to your comment, the statistical analysis between different days after dsGFP injection were indicated by the lowercase (a,b), while the statistical analysis between different days after dsCycB injection were indicated by the capital letters (X, Y). We have revised the Figure 6. We hope it is clear now.

Discussion:

lines 332-334: “In the present study, the highest expressions of Mn-CycB were observed in the testis and ovary of male and female M. nipponense, respectively, which were consistent with the previous studies.” This statement does not represent the exact results, as shown in figure 4A. the transcript levels of female gill are higher than the testis. 

Reply: Yes, the transcript levels of female gill are higher than the testis.  However, in male prawns, the Mn-CycB expression was the highest in the testis. In the sentence of line 332-334, I want to express that testis was observed as the highest expression in all tested tissues from male prawns, while ovary was observed as the highest expression in all tested tissues from female prawns. Thus, our description is correct. We revised it as “In the present study, the highest expressions of Mn-CycB were observed in the testis of males and the ovary of females” at line 285-286, in order to make it clear.

Lines 347-350: here the authors are presenting a misleading conclusion.  No functional genomics experiment was done in the period of P5-PL20. So they only could suggest that this should be tested. Or the authors could perform functional genomics at this stage.

Reply: Thank you for your suggestion. We agreed with your comment. This part has been removed from the manuscript.

Line 360: here again the literature is not touching the most closest Macrobrachium species AG and IAG literature and the IAG-switch.

Reply: Thank you for your reminding. More details of the literatures related to the AG and IAG in Macrobrachium resenbergii have been provided at line 309-316. We hope it meets your demand now.

 The final sentence:

In conclusion, the above evidence in the present study indicated that CycB positively regulated the testis development  spermiogenesis through affecting the expression of IAG in M. nipponense. This study dramatically provides valuable evidences on the calls for further studies of CycB in other crustacean species, as well as promoting the male sexual development.

Reply: Thank you for your suggestion. We have revised it at line 326-330, according to your comment.

Reviewer 2 Report

Comments for Genes-1965610

General comments: the experiment and its presentation in the manuscript was well organized and fine. All are structured in good manner. But some improvement which is crucial need to be added and can be seen as follows:

Tittle

Recheck the sentence, there is a missing word.

The tittle need to be reframed and shorten

Methods

Instead of long narration using sentence, it would be better to add a representative flowchart as figure to explain the time point of sample preparation, and sample collection as well.

Statistical analysis

t-test and ANOVA are different, please specify which data or experiment that analyzed using t-test and which when using ANOVA.

Result

Figure 4.

-        Add the expression level of EIF as normalizer as insert, along with the statistical analysis between the sample. Show if it is expression in all samples are not statistically different.

-        Figure 4A. What does it mean the Yd, Xb, Xd, and so on. There are no explanation in the legend.

-        Figure 4B. See above comment at Fig. 4A, the same cases occur here.

Figure 5.

-        Add the expression level of EIF as normalizer

-        Explain the a and b over the bar in the legend

Figure 6

-        The same as Fig 4 and 5, add the expression level of EIF as normalizer, and show it is not different in all samples

-        Explain what are a and b over the bar. Add the description in the figure legend.

Author Response

General comments: the experiment and its presentation in the manuscript was well organized and fine. All are structured in good manner. But some improvement which is crucial need to be added and can be seen as follows:

Tittle

Recheck the sentence, there is a missing word.

The tittle need to be reframed and shorten

Reply: Thank you for your suggestion. The title has been shorten and revised as “RNAi analysis identified the Cyclin B functions in male reproduction of Macrobrachium nipponense”.

Methods

Instead of long narration using sentence, it would be better to add a representative flowchart as figure to explain the time point of sample preparation, and sample collection as well.

Reply: Thank you for your suggestion. A flowchart has been provided to explain the sample collection in Figure 1. We hope it is clear now.

Statistical analysis

t-test and ANOVA are different, please specify which data or experiment that analyzed using t-test and which when using ANOVA.

 Reply: Thank you for your reminding. The explanation for t-test and ANOVA has been explained at line 178-185.

Result

Figure 4.

Add the expression level of EIF as normalizer as insert, along with the statistical analysis between the sample. Show if it is expression in all samples are not statistically different.

Reply: In the experiment, EIF, just as reference gene, was used to nomalize CT value of the target gene in the caculation of mRNA levels. Generally, the expresion of reference gene was not required to be listed in the standard of caculation via 2−∆∆CT method (Bustin et al., 2009; Livak, Schmittgen, 2001). Meanwhile, in our previous study, the EIF was confirmed to be a suitable and stable reference gene in M. nipponense (Hu et al., 2018). The suitable and stable reference gene was selected from EIF (eukaryotic translation initiation factor 5A), 18S (18S ribosomal RNA), EF-1a (elongation factor-1a), GAPDH (glyceraldehyde-3-phosphate dehydrogenase), TUB (a-tubulin), β-act (β-actin), and RPL18 (Ribosomal protein L18). Authors proved that EIF is the most stable reference gene under different conditions in M. nipponense.

Bustin, S.A., Benes, V., Garson, J.A., Hellemans, J., Huggett, J., Kubista, M., Mueller, R., Nolan, T., Pfaffl, M.W., Shipley, G.L., 2009. The MIQE guidelines: minimum information for publication of quantitative real-time PCR experiments. Clinical Chemistry. 55, 611-622.

Livak, K., Schmittgen, T., 2001. Analysis of relative gene expression data using real-time quantitative PCR and the 2(-Delta Delta C(T)) Method. Methods-A Companion To Methods in Enzymology. 25, 402-408.

Figure 4A. What does it mean the Yd, Xb, Xd, and so on. There are no explanation in the legend.

Figure 4B. See above comment at Fig. 4A, the same cases occur here.

Reply: The explanation of lowercases and captital lettwe has been provided as “Lowercases on the bars indicate expression difference between different tissue samples, analysed by ANOVA test. Capital letters on the bars indicate expression difference in the same tissue between male and female prawns, analysed by independent samples T-test” in the caption of Figure 5 (revised manuscript)

Figure 5.

Add the expression level of EIF as normalizer

Reply: The question has been explained in Figure 4’s comment

Explain the a and b over the bar in the legend

Reply: Thank you for your reminding. We have explained it as “ Lowercases (a, b) on the bars indicate expression difference between different tissue samples, analysed by independent samples T-test” in the caption of Figure 6 (revised manuscript).

Figure 6

The same as Fig 4 and 5, add the expression level of EIF as normalizer, and show it is not different in all samples

Reply: The question has been explained in Figure 4’s comment.

Explain what are a and b over the bar. Add the description in the figure legend.

Reply: Thank you for your reminding. We have explained it as “ Lowercases (a, b) on the bars indicated expression difference between different days after dsGFP injection, while capital letters (X, Y) on the bars indicated expression difference between different days after dsCycB injection, analysed by ANOVA test. * (p < 0.05) and ** (p < 0.01) indicates significant expression difference between the dsGFP and dsCycB treated group at the sample day, analysed by independent samples T-test” in the caption of Figure 7 (revised manuscript).

Round 2

Reviewer 1 Report

the description of the histological results at 3.5 could be still improved. The results (histology) are suggesting effects on sperm formation from the spermatozoa stage. define the different cell types at the different stages and treatments. 

Author Response

The description of the histological results at 3.5 could be still improved. The results (histology) are suggesting effects on sperm formation from the spermatozoa stage. define the different cell types at the different stages and treatments. 

Reply: Thank you for your suggestion. We provided more details on the description of cell types after the treatment of dsGFP and dsCycB in the section 3.5. We hope it meets your demand now.

Reviewer 2 Report

Revized version is ok

Author Response

Revized version is ok。

Reply: Thank you for your agreement.